# Assessing the validity of fecal sampling for characterizing variation in threespine stickleback's gut microbiota

Andreas Härer ⓘ *, Diana J. Rennison

School of Biological Sciences, Department of Ecology, Behavior, & Evolution, University of California San Diego, La Jolla, California, United States of America

* ahaerer@ucsd.edu

## Abstract

The gut microbiota is crucial for many aspects of their hosts' biology, and it has been characterized for many species across the animal kingdom. Yet, we still don't have a good understanding of whether non-lethal sampling can accurately capture the diversity of gut-associated bacterial communities, as estimated from lethal sampling of intestinal tissue. We further lack knowledge on whether non-lethal sampling methods are suitable for detecting gut microbiota shifts associated with changes in environmental factors (e.g., diet). We addressed these questions in threespine stickleback fish, a model system for evolutionary ecology, by comparing bacterial communities from intestinal tissue and feces. Despite some differences in community composition between the two sample types and considerable temporal variation among fecal samples, bacterial communities appear to largely overlap. Further, we detected consistent and significant changes of fecal bacterial communities associated with an experimental diet manipulation. This suggests that fecal sampling can represent an adequate non-lethal method to characterize the gut microbiota of threespine stickleback, but additional studies will be necessary before drawing general conclusions regarding the validity of fecal sampling for gut microbiota studies. To this end, we give recommendations to improve the characterization of the gut microbiota via fecal sampling. Fecal sampling allows studying temporal gut microbiota shifts associated with environmental change at the individual level, which increases opportunities for future experimental gut microbiota research.

## Introduction

The gut microbiota is a complex and dynamic community that is crucial for many aspects of their host's biology, including nutrient metabolism [1, 2]. Bacterial communities associated with the gut have been characterized in thousands of vertebrate host lineages [3, 4]. The type of sample collected for gut microbiota research often varies depending on the study organism. For example, samples are almost exclusively obtained from fecal matter in mammals, which represents a non-invasive way of obtaining a proxy for the microbial communities inhabiting

**Data Availability Statement:** The raw 16S sequencing data (10.6084/m9.figshare.16435185) as well as the R code and the underlying data files

for our analyses (10.6084/m9.figshare.16434801) are deposited on figshare.

**Funding:** This work was supported by funding from the Deutsche Forschungsgemeinschaft (DFG, German Research Foundation) – project number 458274593 to A.H. and from the University of California San Diego to D.J.R.

**Competing interests:** The authors have declared that no competing interests exist.

the gut [3]. In contrast, gut microbiota studies in fish are commonly based on intestinal tissue, which necessitates lethal sampling [5–7]. However, certain research questions might require repeated sampling of the same individuals, e.g., when investigating temporal changes of the gut microbiota [8]. Non-lethal sampling of fecal matter seems an appropriate method to achieve this, but the question of how closely the fecal microbiota resembles that of the host's gut remains contentious. This has been tested only in a limited number of host organisms, including bivalves [9], amphibians [10], mammals [11] and birds [12], producing mixed results. An increasing number of studies are also investigating the gut microbiota of different fish species, and some have directly compared bacterial communities obtained from fecal matter and guts, particularly in the field of aquaculture [e.g., 13–17]. However, the results are mixed as to whether fecal samples can act as a suitable alternative to gut tissue for studying the gut microbiota and prior work has failed to determine whether temporal shifts in response to environmental change or experimental manipulation can be assessed using these methods or whether repeated sampling better predicts overall gut microbiota diversity. Thus, the questions of how stable microbial communities associated with fecal matter are, and how many fecal samples might be needed to capture the bacterial diversity found in the gut, remain unanswered.

Threespine stickleback fish (*Gasterosteus aculeatus*; hereafter referred to as stickleback) are widespread across the Northern hemisphere and have become a model system in the fields of behavioral ecology, evolutionary biology, and adaptation genomics [18–20]. There is also increasing interest in stickleback's gut microbiota. Preliminary work has begun to characterize the contributions of host-associated (e.g., ecology, genotype, sex) and environmental factors to variation in gut microbial diversity, within [21–23] and across populations [24–26]. These findings suggest that ecological and evolutionary processes, to some degree, govern assembly of stickleback's gut bacterial communities. Stickleback are also considered a textbook example of parallel evolution [19], and a recent study by Rennison et al. found that repeated changes in trophic ecology are associated with parallel shifts of the gut microbiota [24]. This study offered first insights into the question of how predictable gut microbiota changes are when stickleback adapt to novel ecological niches. Yet, we argue that controlled laboratory manipulation experiments are necessary to further assess how host organisms and their gut microbiota interact when different populations are exposed to novel diets, and how this might shape the ecology and evolution of stickleback. However, many of these experiments would require replicate sampling of the same individuals, which necessitates non-lethal methods, such as the sampling of fecal matter.

Here, we tested whether fecal sampling represents an adequate technique for characterizing gut bacterial communities of stickleback fish. To this end, we performed two experiments to explore the validity of fecal sampling: first, we determined the variability of the bacterial communities associated with fecal matter over a time span of 7–12 days and compared these to the gut microbiota characterized from whole gut extractions of the same fish. Second, we tested whether a change in diet leads to a detectable and consistent change in bacterial communities characterized through replicate fecal sampling. The results obtained in this study will help to evaluate whether fecal sampling can be used to study temporal gut bacterial community dynamics in stickleback, and likely other fish species.

## Materials and methods

### Experimental set-up and data collection

Experiments were approved by the University of California San Diego Institutional Animal Care and Use Committee (protocol number S20031) and were performed in accordance with

institutional guidelines. We studied two different stickleback populations from Vancouver Island, British Columbia, Canada: a freshwater population from Gosling Lake and a marine population from the Sayward Estuary. The fish were brought into our research facility at the University of California, San Diego campus, as fertilized crosses from lab-reared fish. Eggs were hatched into separate static tanks and raised for one year under standardized abiotic conditions, which were maintained during the experiments (temperature: 16˚C, salinity: 3.5 ppt, pH: 7.8). All tanks were filled with dechlorinated city water processed through a reverse osmosis system. Tanks were oxygenated with air stones and in-tank filters. All fish were fed brine shrimp once a day prior to the start of each experiment. We performed two experiments during which fish were fed *ad libitum* with standardized amounts of diet (either brine shrimp or blood worms) once a day. At the beginning of each experiment, fish were introduced into fine-meshed net breeders in two previously empty static tanks, with an acclimation period of 24 hours before the start of the experiment. All fish used for the experiments were selected randomly from their source populations. Each tank had five fine-meshed net breeders, with one fish in each net breeder to allow collection of fecal samples from individual fish, and all fish used for the two experiments (except for Sayward 3) were kept in the same experimental tank. Measurements with digital calipers showed that fish ranged in standard length from 4.5–6 cm.

In the first experiment, we fed fish from Gosling Lake (n = 4) and the Sayward Estuary (n = 3) brine shrimp once a day and collected fecal samples over a period of 7–12 days until we had six fecal samples per individual (S1 Table). The sample size for Sayward population was smaller because PCR amplification failed for the gut sample of one individual, hence, it was excluded from our analyses. Once the experiment was completed for a fish, a new fish was introduced into the empty net breeder. Net breeders were checked for the presence of feces twice daily; in the morning right before feeding and in the afternoon (approximately 6 hours after feeding). Fecal samples were collected using disposable plastic pipettes, and separate pipettes were used to collect samples from each fish. Residual tank water was carefully removed by pipetting, and the samples were stored at -80˚C until DNA extraction. After the final fecal sampling (either 6 hours or 24 hours after feeding depending on the time of day the last fecal sample was collected), we sacrificed fish using an overdose of MS-222. Fish were rinsed with EtOH, whole guts were dissected using sterile equipment, as commonly done in gut microbiota studies in stickleback [e.g., 22, 24]. Guts were then gently squeezed to remove the majority of gut contents; however, we would like to emphasize that this method does not ensure the complete removal of gut contents as small amounts of fecal residues might persist in the gut. Samples were stored at -80˚C until DNA extraction.

For the second experiment, only fish from Gosling Lake (n = 5) were used and the experiment was conducted simultaneously in one tank for all five fish. For this experiment, fish were fed brine shrimp once a day for three days, and two fecal samples per fish were collected. Then, their diet was switched to blood worms and fecal samples were collected starting one day post diet change. This experiment lasted a total of 13 days, with 6–12 samples collected per fish (S1 Table). Before the diet change, fecal samples were collected in the afternoon. After the diet change, samples were collected either in the morning or in the afternoon (S1 Table). We found that time of day did not affect alpha diversity (Wilcoxon rank-sum test, $P > 0.05$ for all measures), nor bacterial community composition (beta diversity, measured as Bray-Curtis dissimilarity; PERMANOVA, $F = 1.81$, $r^2 = 0.06$, $P = 0.106$). This suggests that the time a fecal sample has been in the tank water does not significantly affect the bacterial community, at least over a period of $< 24$ hours. For one individual (Gosling 7), we collected samples in the morning and in the afternoon over four days (and additional samples were collected before and after that), but only the morning samples were kept for illustrating temporal changes in the proportions of bacterial phyla and in alpha diversity (leading to 4–6 samples/individual

after diet change as depicted in Fig 4). Additionally, we collected two samples of each diet item (brine shrimp, blood worms), which allowed for comparisons of bacterial communities from fecal and gut samples with those associated with the diet items. Brine shrimp were reared in separate tanks in the research facility at UCSD, and residual water was removed prior to sampling. Blood worms were kept at -20°C before feeding and thawed before sampling. Thus, both diet reference samples did not contain experimental tank water. After sampling, diet items were stored at -80°C until DNA extraction.

DNA was extracted from diet items, fecal and gut samples using the QIAGEN PowerSoil Pro Kit according to the manufacturer's protocol (Qiagen, Hilden, Germany). DNA extraction and PCR amplification were done under sterile conditions in a laminar flow hood to minimize contamination risk. DNA concentrations were measured on a Qubit 4 Fluorometer (Thermo Fisher Scientific, Waltham, MA). Negative controls of sterile $H_2O$ were included during DNA extraction and PCR amplification, which did not yield detectable DNA concentrations. We amplified the V4 region of the 16S rRNA gene using barcoded 515F and 806R primer (see the protocol published by Kozich et al. for primer sequences: https://github.com/SchlossLab/ MiSeq_WetLab_SOP/blob/master/MiSeq_WetLab_SOP.md). All PCR amplifications were performed in triplicates with a 10 µl reaction volume using Platinum II Hot Start PCR Master Mix (Thermo Fisher Scientific), and replicates were subsequently pooled. The PCR protocol consisted of an initial denaturation step for 60 s at 98°C, 35 amplification cycles with 10 s at 98°C, 20 s at 56°C and 60 s at 72°C, and a final elongation at 72°C for 10 min. Gel electrophoresis (2% agarose gel) was performed for all samples to visually check for amplification specificity. DNA concentrations of pooled PCR products were again measured on a Qubit 4 Fluorometer, and samples were pooled in an equimolar manner to construct a single library. This library was shipped to the UC Davis Genome Center where it was purified by bead clean-up and run on a Bioanalzyer to check DNA quality. The final library was sequenced on one lane of the Illumina MiSeq 600 (PE300) platform. The raw sequencing data have been deposited on figshare, see Data Accessibility Statement and [27]. The methods and results of our study are reported in accordance with ARRIVE guidelines [28].

## Gut microbiota analysis

Across all the samples included in this study, we obtained a total of 4,312,292 raw sequencing reads (median: 47,067 reads/sample; mean: 46,872 reads/sample; SD = 16474; S1 Table). Reads were imported into the open-source bioinformatics pipeline Quantitative Insights Into Microbial Ecology (QIIME2) [29]. Unfortunately, we only recovered low sequencing depths for some samples, particularly for gut samples (S1 Table), and merging of forward and reverse reads further decreased these numbers due to filtering of reads during the merging step. Hence, we decided to use 250 bp of only the forward reads since sequence quality was much higher for forward reads than for reverse reads. In order to obtain amplicon sequencing variants (ASVs), we used the QIIME2 plugin *dada2* to check the quality of our sequence data, correct sequencing reads, and filter chimeric sequences [30]. We constructed a phylogenetic tree of the bacterial lineages present in our samples with FastTree 2.1.3 [31] as implemented in QIIME2, which is necessary to calculate phylogenetic distance matrices for the two UniFrac metrics. We assigned bacterial taxonomy against the SILVA 132 ribosomal RNA (rRNA) database at a 99% similarity threshold [32]. We then filtered ASVs that (i) only occurred in one sample, (ii) could not be assigned below the phylum level, or (iii) belonged to either chloroplasts, mitochondria, Cyanobacteria, or archaea. We decided to filter Cyanobacteria because they are not expected to be functional or permanent members of the gut microbiota since they are mainly free-living photosynthetic bacteria. Further, their relative abundance was very low

(only on average 1.5% of sequencing reads) and Cyanobacteria were not detected in 60 out of 94 samples. Our results were qualitatively similar, whether Cyanobacteria were removed or not and, thus, this filtering step did not affect our alpha and beta diversity results and therefore our conclusions. We performed all subsequent statistical analyses using two slightly different data sets: (i) we filtered ASVs that represented more than 10% of the bacterial communities of the diet items, to make sure that the fecal bacterial communities do not merely resemble the microbes present in the diet, (ii) we did not filter any diet-derived bacteria (see S1 Table for filtered read numbers and sample sizes for each experiment). The filtering of diet-associated ASVs led to the exclusion of only five ASVs belonging to the bacterial families Carnobacteriaceae, Alteromonadaceae, Vibrionaceae, and Cyclobacteriaceae. We would like to highlight that the results were qualitatively similar between the two analyses. Hence, we present statistics and figures primarily from the analysis including diet filtering and highlight when results deviated between the two analyses (all data files and R code used for all statistical analyses for the two types of data can be found in the figshare repository, see Data Accessibility statement for details). Prior to the diet filtering step, we tested the proportions of fecal bacteria that were present only in the gut of the same individual, only in the diet, in both of them or in neither of them (S1 Fig). We further determined the proportion of ASVs from intestinal samples that were also captured in fecal samples of the same individual (Fig 2C and S2 Fig). Only bacterial phyla with a mean proportion of more than 0.2% are shown in the taxonomic bar plots. Data was rarefied to 4238 and 4275 sequencing reads for the first (comparison of guts and feces) and second (diet change) experiment, respectively. This represents the sampling depths we used for all alpha and beta diversity analyses of the bacterial communities. Rarefaction analyses showed that these sampling depths were sufficient to capture substantial proportions of bacterial alpha diversity, although we acknowledge that rarefaction curves indicate a sampling depth of 10,000 reads would have been necessary to capture the full bacterial diversity (S3 Fig).

We used three metrics to describe alpha diversity (amplicon sequence variant (ASV) richness, Shannon diversity, and Pielou's evenness) and three metrics to describe beta diversity (non-phylogenetic: Bray-Curtis dissimilarity, phylogenetic: unweighted and weighted UniFrac) [33, 34]. All subsequent statistical analyses were done in R v4.0.5 [35]. Results were consistent across the different beta diversity metrics, we report all statistics in the main text, but we only visualize Bray-Curtis dissimilarity based on principal coordinate analysis. Differences in alpha diversity measures between diet types and populations were tested with non-parametric Wilcoxon rank-sum tests [36], and Kruskal-Wallis tests were used for comparisons among individuals, as implemented in the R stats package [37]. Spearman's rank correlation coefficients were calculated to test for correlations between alpha diversity and the overlap between bacterial communities from fecal samples and gut tissue (S4 Fig). When comparing Bray-Curtis dissimilarity among fecal samples and between fecal and gut samples of the same individual, we performed Wilcoxon rank-sum tests and adjusted p values for multiple comparisons using the Bonferroni correction (S5 Fig). To test for differences in bacterial communities among host populations and sample types (beta diversity), we applied Permutational Multivariate Analysis of Variance (PERMANOVA) [38] based on distance matrices, using the *adonis2* function of the R vegan package. We controlled for repeated sampling within individuals by constraining permutations with the strata option. Statistical significance was determined at the 0.05 level. We further tested for differential abundance of bacterial phyla and genera between guts and feces in the first experiment (S2 and S3 Tables) and between fecal samples of fish fed either brine shrimp or bloodworms at the time point of sampling in the second experiment (S4 and S5 Tables) using ANCOM as implemented in QIIME2.

## Results

### Comparison of repeated fecal sampling and lethal gut sampling

The relative abundance of bacterial phyla associated with feces (mean values across temporal samples) and the gut varied substantially across individuals (Fig 1A). The bacterial communities, in both sample types, largely consisted of Proteobacteria (mean feces: 39.2–77.4%, guts: 6.2–84.2%) and Actinobacteria (mean feces: 22.1–59.2%, guts:7.9–89.7%; Fig 1A). Bacteroidetes were much more abundant in fish originating from Sayward (2.1–22.7%) compared to fish from Gosling (0–2%). The proportion of the fecal bacterial community that was also found in the gut microbiota of the same individual (either exclusively or in the gut microbiota and in the diet) was high for most individuals (median: 90.3%), and although there was considerable variation, only 3 out of 42 fecal samples showed proportions of less than 50% (22–99.9%; S1 Fig).

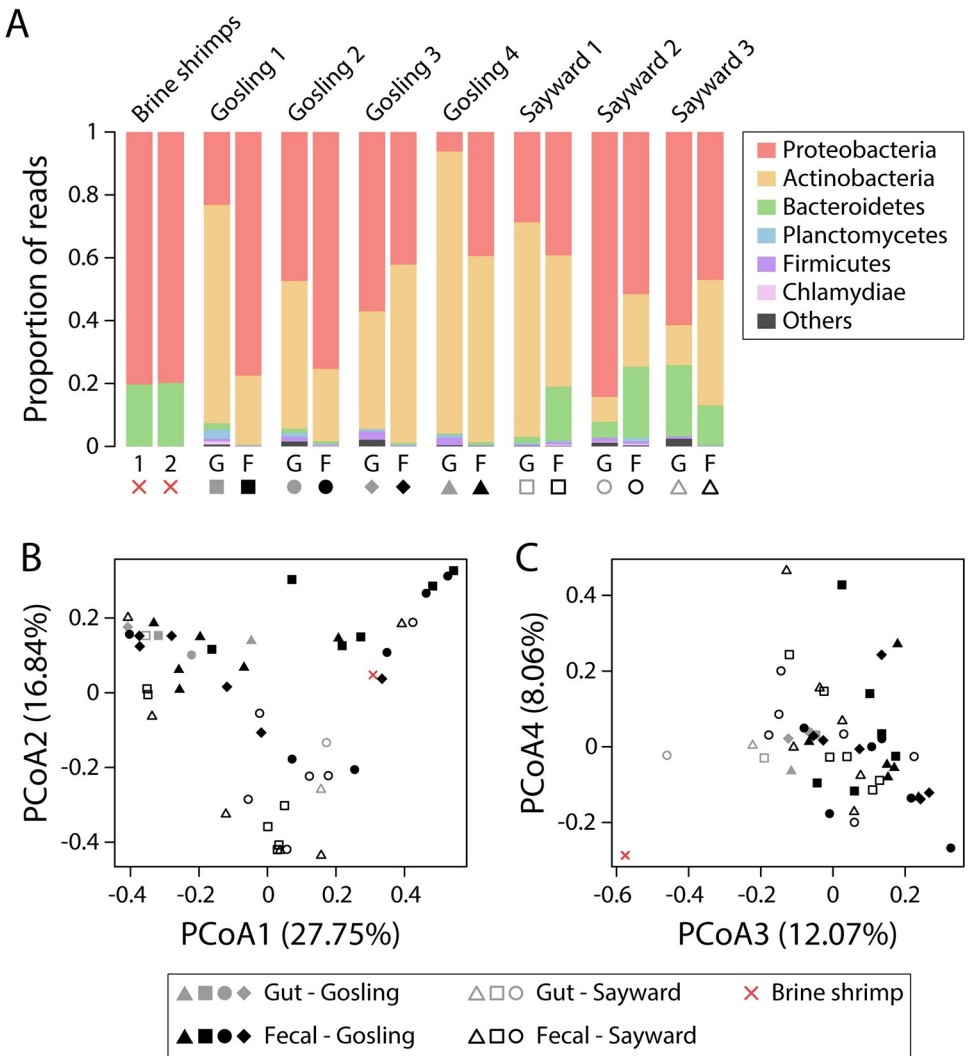

**Fig 1.** Taxa bar plots showing that microbial communities largely consisted of Proteobacteria, Actinobacteria and Bacteroidetes for diet items (brine shrimp), feces (mean values across temporal samples) and guts (A). Bacterial ASVs that represented more than 10% of brine shrimp's bacterial communities were filtered from fecal and gut samples. There was no obvious clustering of fecal and gut samples along the first four PCoA axes (B & C), but populations appeared to separate along PCoA axis2 (B) and PCoA axis 3 separated diet items from fecal and gut samples (C).

Patterns of beta diversity were consistent across metrics and did not depend on whether diet-derived bacteria were filtered or not. Here, we report test statistics after diet filtering, unless stated otherwise. When estimating Bray-Curtis dissimilarity, bacterial communities significantly differed between the two source populations (PERMANOVA, $F = 5.97$, $r^2 = 0.11$, $P = 0.005$) but also between sample types (feces vs. guts) ($F = 3.29$, $r^2 = 0.06$, $P = 0.007$). Similar results were obtained using unweighted (population: $F = 4.39$, $r^2 = 0.08$, $P = 0.002$; sample type: $F = 3.61$, $r^2 = 0.07$, $P = 0.003$) and weighted UniFrac (population: $F = 5.98$, $r^2 = 0.11$, $P = 0.028$; sample type: $F = 2.71$, $r^2 = 0.05$, $P = 0.039$). Differentiation between populations could be seen along PCoA axis 2, but there was no apparent clustering by individual or between fecal and gut samples along the first four PCoA axes (based on Bray-Curtis dissimilarity; Fig 1B and 1C). The bacterial communities of diet items (brine shrimp) did not separate from the fecal and gut samples along PCoA axes 1 and 2, even after filtering bacteria that were highly abundant in brine shrimp (relative abundance of >10%) from fecal and gut samples (Fig 1B). Yet, brine shrimp were clearly differentiated from the fecal and gut samples along PCoA axis 4 (Fig 1C). When filtering diet-derived bacteria, pairwise Bray-Curtis dissimilarity values among samples of the same individual did not differ among fecal samples or between fecal and gut samples for all but one individual (Sayward 2; S5 Fig). For unweighted and weighted UniFrac, only two individuals (Gosling 3 & 4) and one individual (Gosling 1) showed significant differences, respectively. Without filtering diet-derived bacteria, results differed for pairwise Bray-Curtis dissimilarities, with significant differences detected in two individuals (Sayward 2 & Gosling 3) and evidence for significant differences for an additional individual (Sayward 2) based on unweighted UniFrac. In all these cases where we detected significant differences, distances were larger between fecal and gut samples than for comparisons among fecal samples (e.g., Sayward 2 in S5 Fig).

There was pronounced variation across temporal fecal samples from the same individual, both in terms of the taxonomic composition and alpha diversity of bacterial communities (Fig 2). For example, the proportion of Proteobacteria within one individual (Sayward 3) ranged from 1.4–98.8% across time points, and substantial differences in relative abundance of bacterial phyla were observed across all individuals (Fig 2A). Alpha diversity, measured as ASV richness, Shannon diversity, and Pielou's evenness, were relatively stable across temporal samples in some individuals (e.g., Gosling 3 & 4) but fluctuated considerably in others (e.g., Gosling 2 & Sayward 1). In several individuals, no alpha diversity estimates from fecal samples matched those from the gut samples (dashed lines in Fig 2B), indicating differences in the bacterial diversity recovered from the two different sample types. Yet, alpha diversity did not differ significantly between populations (Wilcoxon rank-sum test, $P > 0.05$ for all measures), between feces and guts (Wilcoxon rank-sum test, $P > 0.05$ for all measures) or among individuals (Kruskal-Wallis test, $P > 0.05$ for all measures), this was consistent whether diet-derived bacteria were filtered or not. For all three alpha diversity measures, we detected significant negative correlations with the overlap between bacterial communities of fecal samples and the gut tissue based on Spearman's rank correlation coefficient (number of ASVs: $\rho = -0.597$, $P < 0.001$; Shannon diversity: $\rho = -0.606$, $P < 0.001$; Pielou's evenness: $\rho = -0.349$, $P = 0.024$) (S4 Fig). The proportion of bacterial ASVs associated with the intestinal tissue that were also detected in single fecal samples of the same individual differed substantially within and across individuals, ranging from 21.1–81% (median: 56%; S2 Fig). The proportion of identified ASVs increased with the number of fecal samples (Fig 2C); pooling of all fecal samples from an individual recovered a large proportion of the intestinal bacterial community (75–93.5%; median: 82%). Only one out of 14 phyla (Deinococcus-Thermus) showed a significant difference in relative abundance between feces and guts (S2 Table). On the genus level, out of a total of 179 genera only Rheinheimera differed significantly when diet-derived bacteria were not filtered

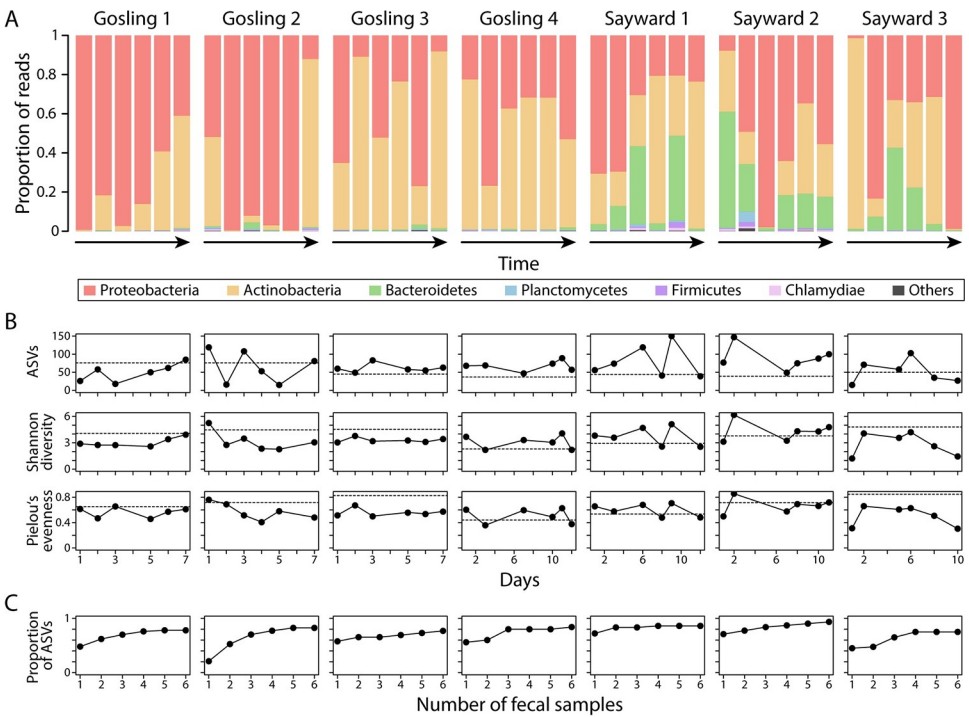

**Fig 2.** Repeated sampling revealed considerable temporal variation in fecal bacterial community composition of the same individual (A). Alpha diversity, measured as ASV richness, Shannon diversity, and Pielou's evenness were rather consistent for some samples (e.g., Gosling 3 & 4) but differed substantially for others (e.g., Gosling 2 & Sayward 1) (B). The horizontal dashed line indicates the alpha diversity for the gut sample of each individual (B). The proportion of bacterial AVSs from the intestinal tissue that were also captured in feces increased with the number of fecal samples (C).

(S3 Table). The genus Rheinheimera was highly abundant in brine shrimps, and after diet filtering, four genera (Acinetobacter, Gemmobacter, Meiothermus and Paraburkholderia) differed significantly. Notably, these four genera also showed the strongest support for differential abundance beside Rheinheimera before diet filtering (S3 Table). Hence, results were largely consistent across the two types of analyses.

## Detection of diet-associated changes in bacterial communities using fecal sampling

We next tested whether a change in diet from brine shrimp to blood worms would be reflected by changes in fecal bacterial communities. For Bray-Curtis dissimilarity, fecal samples differed significantly between diet types (PERMANOVA, $F = 9.13$, $r^2 = 0.2$, $P = 0.001$), the same pattern was observed for unweighted ($F = 17.34$, $r^2 = 0.32$, $P = 0.001$) and weighted UniFrac ($F = 7.29$, $r^2 = 0.16$, $P = 0.002$), and patterns were consistent whether diet-derived bacteria were filtered or not. These results were also reflected by a clustering of fecal samples by diet along PCoA axis 2 in the same direction as the diet items (Fig 3A), whereas PCoA 3 separated diet items from fecal samples based on Bray-Curtis dissimilarity (Fig 3B). Alpha diversity was significantly higher when fish were fed blood worms compared to brine shrimp, both for ASV richness (Wilcoxon rank-sum test, $P < 0.001$; Fig 3C) and Shannon diversity ($P = 0.001$; Fig 3D) but not for Pielou's evenness ($P > 0.05$; Fig 3E). The fecal bacterial communities of fish fed brine shrimp largely consisted of Proteobacteria (34.2–99.5%), but also Actinobacteria were highly abundant in some samples (0.4–53%; Fig 4A). After changing diet to blood worms

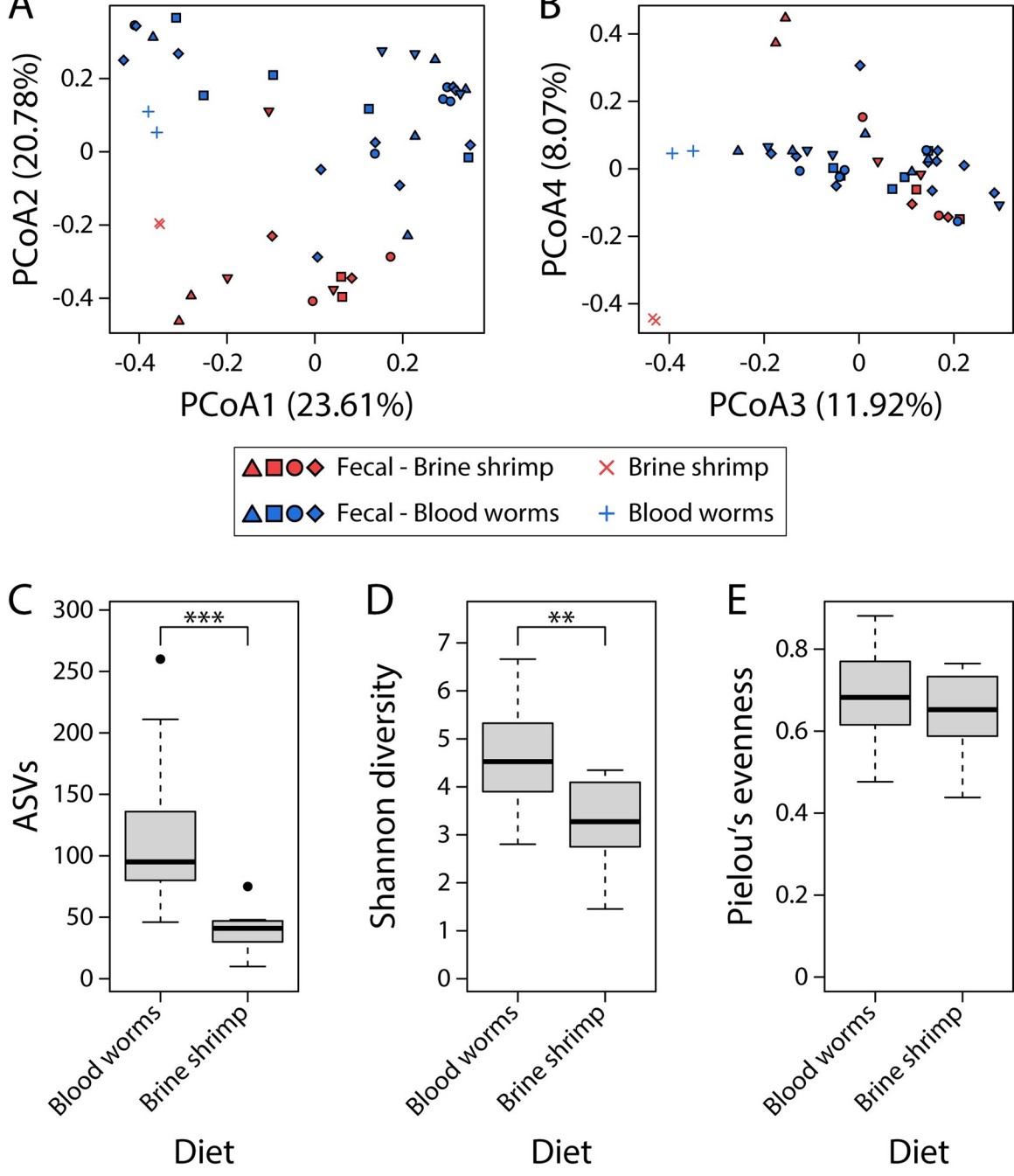

**Fig 3.** Diet strongly affected fecal bacterial communities and separated samples along PCoA axis 2 (A); diet items separated from fecal samples along PCoA axis 3 (B). Alpha diversity measures were significantly higher for fish fed blood worms compared to fish fed brine shrimp for ASV richness and Shannon diversity, but not for Pielou's evenness, based on Wilcoxon rank-sum tests (C-E). **P < 0.01, ***P < 0.001. The shapes of the filled symbols represent samples from different individuals and the different colors refer to the diet that fish were fed at the time of sampling.

(indicated by the vertical dashed lines in Fig 4), Firmicutes drastically increased in abundance, constituting up to 68.3% of bacterial communities, whereas they represented less than 1% in fish fed brine shrimp (Fig 4A). After the diet change, alpha diversity values initially strongly increased and then decreased over time (except for Gosling 6, which showed the highest alpha

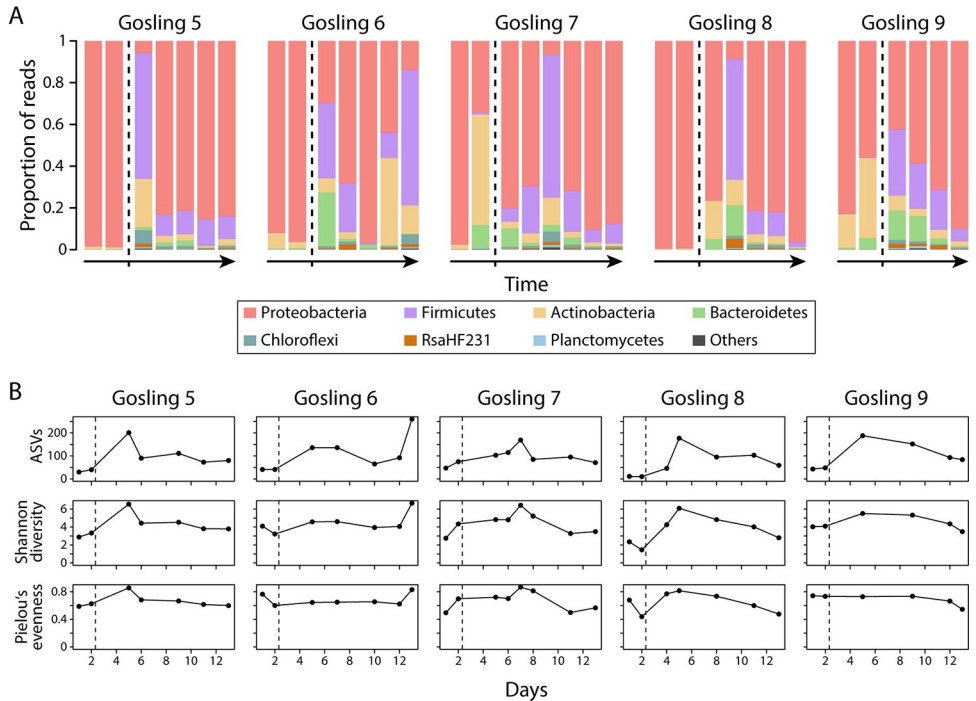

**Fig 4.** A change in diet from brine shrimp (first two data points for each individual, left of vertical dashed line) to blood worms (data points right of vertical dashed line) induced substantial changes in fecal bacterial communities in all five individuals (A). Alpha diversity generally increased after the diet change, decreased over time (except for Gosling 6) but appeared to remain higher than before the diet change for some host individuals (B).

diversity value at the last time point; Fig 4B), and some fish maintained higher alpha diversity than before the diet change (but note that results slightly differed across alpha diversity metrics). Three phyla (Chloroflexi, Firmicutes and RsaHF231) out of a total of twelve were differentially abundant between the two diet treatments independent of diet-derived bacterial filtering (S4 Table), and an additional phylum (Planctomycetes) was differentially abundant after filtering diet-derived bacteria. On the genus level, 18 out of 238 genera were differentially abundant between diet treatments independent of diet-derived bacterial filtering (S5 Table). Two additional genera (Carnobacterium and Vibrio) differed between diet treatments without diet-derived bacterial filtering; these two genera were also highly abundant in diet items and, hence, were not included in the diet-filtered data set.

## Discussion

The bacterial communities of feces and intestinal tissue have been compared directly in host lineages across the animal kingdom [e.g., 9–12] including teleost fishes [e.g., 15–17, 39, 40]. However, previous studies are commonly restricted to a single time point, but for many experiments it is useful or even imperative to collect repeated samples and track gut microbiota changes over time on the individual level. This is not possible if lethal sampling is necessary to characterize the gut microbiota. Thus, it is important to determine for a given study system to what extent the bacterial communities obtained from non-lethal fecal sampling differ from lethal intestinal sampling, if bacterial communities associated with fecal matter are stable over time and whether fecal samples can recover most taxa identified using whole gut extractions. Such analyses will allow assessment of whether fecal sampling represents an adequate tool for studying gut bacterial communities [e.g., 9] and for characterizing temporal dynamics. To this

end, we characterized fecal bacterial communities in stickleback fish, an important model organism in evolutionary ecology, whose gut microbiota has been the focus of several studies over the last years [22, 24–26].

Overall, we found somewhat conflicting evidence regarding the validity of using fecal samples as a proxy for the gut microbiota in stickleback. The fact that we detected significant differences between sample types (feces vs. guts) and variation among fecal samples of the same host individual generally argues against using fecal samples when interested in the composition of the gut microbiota. However, our results still suggest that fecal sampling could be used for studying gut microbiota of stickleback and we present our arguments here. First, host population explained a larger proportion of the variance in microbiota composition than sample type in our first experiment (10.7% vs. 5.9%, based on PERMANOVA for Bray-Curtis dissimilarity), suggesting that we were able to detect population-specific signals (Fig 1B). Notably, our relatively small sample size (n = 7) was sufficient to capture microbiota structuring associated with these factors, although we acknowledge that larger sample sizes are necessary to draw more robust conclusions [41] and to increase the chance of detecting biologically relevant signals in the data. This is particularly due to the substantial variation in microbial community composition among fecal samples, between feces and guts, and among host organisms. Second, on the individual level, we detected similar levels of microbiota variation among fecal samples, and between fecal and gut samples (except for Sayward 2, S5 Fig). These within-host results are in contrast to the among-host results that showed significant differences between sample types, but overall indicate that the two sample types harbor similar bacterial communities at the individual host level. Both gut and fecal bacterial communities were further dominated by three phyla: Proteobacteria, Actinobacteria and Bacteroidetes (Figs 1 and 2), and Firmicutes were highly abundant in most fecal samples of fish that were fed blood worms (Fig 4A). These findings are consistent with previous studies [23, 24, 26], providing further support that fecal samples are largely representative of sticklebacks' gut microbiota. Third, only a few bacterial groups (investigated on the genus and phylum level) differed in abundance between fecal and gut samples, while many more bacterial groups were significantly different in their abundance between fecal samples of fish before and after the diet change (S2–S5 Tables). These results suggest that a change in diet can induce a stronger effect on microbial community composition than sample type (feces vs. guts), thereby providing further evidence that fecal sampling can represent an appropriate and sufficient tool for studying temporal gut microbiota shifts in response to experimental manipulation. Lastly, despite some temporal variation in taxonomic composition (Fig 2A) and alpha diversity (Fig 2B), we found that a large proportion (median: 90.3%) of fecal bacterial communities were also found in the gut of the same individual (S1 Fig). Taken together, it currently remains somewhat contentious whether fecal sampling represents an adequate tool for characterizing the stickleback gut microbiota. Yet, we present several lines of evidence which suggest that fecal sampling captures most bacterial diversity of the host's gut microbiota and we argue that it can potentially represent an adequate method for studying gut microbiota dynamics. At the same time, we acknowledge that our study comes with certain limitations such as high variation among temporal fecal samples and small sample size [41]. To this end, we provide specific suggestions to improve non-lethal sampling techniques, to obtain more reliable and reproducible estimates of gut microbial communities in the future.

Our results revealed some notable patterns that have implications for future studies, and therefore warrant further discussion. The proportions of fecal bacterial communities shared with intestinal samples varied substantially across individuals (22–99.9%), but also across time points within the same individual (S1 Fig). One explanation could be that for some fecal samples, we captured a large proportion of bacteria associated either with diet or the tank water.

While we aimed to carefully remove residual tank water associated with the fecal samples, we cannot exclude the possibility that the actual proportion of tank water relative to fecal matter varied between samples and might have contributed to the observed temporal variation in microbial communities. This explanation appears to be in line with our findings that alpha diversity of fecal samples was negatively correlated with the proportion of bacteria shared with the gut tissue of the same individual (S4 Fig). Accordingly, alpha diversity of the gut sample was lower than that of the fecal samples for some individuals (see ASV numbers for Gosling 3 & 4, Sayward 1 & 2 in Fig 2B), and we also detected strong temporal variation in the overlap of bacterial communities between feces and the gut (e.g., Gosling 3 and Sayward 2 in S1 Fig). One could also hypothesize that fecal samples collected temporally closer to the gut tissue sampling could show stronger overlap, but we ruled out this possibility as there was no significant correlation between these two variables (Spearman's rank correlation, $\rho$ = -0.06, $P$ = 0.69). However, we would like to acknowledge that the time since last feeding might have differed across host individuals for the gut sampling since we did not record whether the fish were sacrificed in the morning (24 hours after feeding) or in the afternoon (approximately 6 hours after feeding) which might affect the composition of the gut microbiota [16].

To provide a more accurate picture of the host's gut microbiota, it is advisable to collect comprehensive information on bacterial communities of different sources (i.e., host gut, diet, or environment). It should be noted that we filtered bacterial ASVs that were highly abundant in diet items, but we regrettably did not characterize the bacterial communities of tank water in this study. Yet, previous studies on wild-caught fish have shown that the gut microbiota strongly differs from the bacterial community of the aquatic environment [5, 6, 42] and that the host's gut microbiota is more strongly associated with the bacterial community of prey items [25]. We do not know whether this holds true for fecal samples, so this should be addressed in future studies. However, our main goal was not to identify the source of ASVs found in fecal samples, but rather to assess patterns of variation across samples. The generally large overlap in bacterial communities between sample types gives us confidence that we largely captured gut microbiota diversity in the fecal samples (S1 and S2 Figs). We would also like to emphasize that microbiota diversity and composition of fecal samples were not significantly affected by sampling time point (i.e., morning or afternoon), indicating that samples could be collected within a time span of up to 24 hours after feeding without substantial changes to bacterial communities associated with feces (see Methods section for more details). However, our study design did not enable us to measure the exact length of time a fecal sample has spent in the water. This, in turn, limits the inferences we can make regarding the stability of bacterial communities and future studies could aim to test for the effect of time since defecation.

Gut microbiota composition can fluctuate from day to day [43], which could explain variation in the degree of overlap between fecal and gut bacterial communities. In this case, we might expect among-individual differences in the proportion of fecal bacteria that cannot be assigned to the gut, and rather are likely from diet items or other sources (e.g., tank water), which we indeed observed (e.g., Gosling 1 vs. Gosling 4; S1 Fig). Along this line, there is the possibility that the order in which fish were introduced to the experimental tank might affect temporal variation of fecal microbiotas since the bacterial community of the water could be expected to change over time. As mentioned above, we recommend collecting data on bacterial communities of the environment (e.g., tank water) in future studies to obtain a more comprehensive picture of temporal changes in microbial communities. Nonetheless, assuming daily fluctuations in gut microbiota composition, the degree of overlap between fecal sample and gut tissue appeared to be fairly robust in our study. Further, there was substantial variation in the proportion of bacterial ASVs associated with intestinal tissue (which could be regarded as a proxy for the "true" gut microbiota diversity), that were also captured in fecal samples within

and across individuals (S2 Fig). Pooling multiple fecal samples consistently increased the proportion of intestinal ASVs recovered (Fig 2C). Taken together, in order to control for temporal variation and increase the proportion of gut microbiota diversity captured, we suggest collection of temporal replicates of fecal samples (Fig 2C).

A change in diet led to significant and consistent shifts of the fecal bacterial communities, even after filtering diet-associated bacteria which indicates that the observed differences are not merely driven by differential acquisition of microbes associated with diet items. (Figs 3 and 4). These results are consistent with previous studies showing that gut microbiota composition is affected by changes in diet based on fecal sampling in humans [44, 45] and mice [43], and based on gut tissue sampling in stickleback [22, 23]. Hence, we argue that fecal sampling enables studying temporal gut microbiota changes on the individual level in stickleback, and potentially other fish species. Yet, we would like to emphasize that the study design should be devised according to the biology of the species investigated. For example, removing individuals from their social context might induce stress, which could in turn affect their microbiota. In such cases, either using settings that allow individuals to interact (e.g., separation by permeable/invisible barriers) or separating individuals only temporarily for sample collection could alleviate stress-related microbiota changes. Here we provide suggestions for future lines of research in teleost fishes, using stickleback, an important model system for evolutionary ecology [18], as an example. Fecal sampling would allow tests of whether the gut microbiota of independent populations adapted to a certain diet responds similarly when environmental factors change (e.g., exposure to a novel diet). Further, if populations differ in their dietary niche width, as seen in stickleback from lakes of varying sizes [46], it could be hypothesized that fish from populations feeding on a broader range of diets show more rapid adjustments of their gut microbiota to a novel diet compared to fish from highly specialized populations. Since microbial communities differ in their capacity to harvest energy from the diet [1], the rate and magnitude to which the gut microbiota can shift in response to a novel diet will affect the host's physiological condition. Another question that could be addressed using fecal sampling is how the presence of other intra- or interspecific organisms can alter the gut microbiota of a focal organism. It has been shown that interhost dispersal affects gut microbiota composition [47], but the temporal dynamics of such changes remain unknown. Such experiments would provide novel insights into how differences in gut microbiota plasticity might affect acclimation and adaptation to novel ecological niches [48], and potentially improve our understanding of how and why some organisms adapt to novel environments while others do not.

After the diet change, we also observed a temporary increase in alpha diversity (Fig 4B), which might be the result of a broader range of nutrients present in the transition phase between the two diets or by the exposure to a higher diversity of diet-associated bacteria. Notably, a previous study in stickleback found the opposite pattern; a mixed diet led to lower gut microbiota diversity compared to a pure diet [22]. However, in our study the increase in alpha diversity was detected within days after changing diet, whereas Bolnick et al., exposed fish to a certain diet for one month before characterizing their gut microbiota, which might explain the varying results. It is apparent that additional research is necessary to infer general patterns of temporal gut microbiota shifts associated with changes in environmental conditions (e.g., diet). Our findings also highlight the utility of repeated sampling for understanding the temporal scale of these shifts in bacterial composition, as even transient or relatively brief shifts in diversity and composition could affect host fitness. We would also like to note that a previous study in threespine stickleback found that diet-induced changes of the gut microbiota varied across host sex [23], which could contribute to the observed variation in gut microbiota shifts among individuals. While scoring sex-specific differences in gut microbiota changes associated with a change in diet was not within the scope of our study, this should be considered for future studies.

In sum, we detected differences between fecal and gut bacterial communities and our study also comes with certain limitations that should be addressed in future studies before drawing general conclusions regarding the validity of fecal sampling for stickleback gut microbiota studies. Yet, we argue that our results suggest that fecal sampling might represent an adequate non-lethal technique to characterize the gut microbiota and might further allow studying temporal gut microbiota dynamics in stickleback, and likely other teleost fish, because (i) there was a large overlap between bacterial communities obtained from feces and the gut, (ii) fecal samples enabled characterizing significant differences in bacterial communities between populations and (iii) fecal samples revealed significant changes in bacterial communities due to an experimental change in diet (even after filtering diet-associated bacteria). Temporal non-lethal sampling of bacterial communities will enable tracking of gut microbiota shifts on the individual level, e.g., in response to novel environmental conditions such as different diets, thereby opening up intriguing possibilities for future lines of gut microbiota research, not only in threespine stickleback but also in many other fish species.

## Supporting information

**S1 Fig. Bar plots showing that large proportions (median: 90.3%) of most fecal bacterial communities were shared with the gut microbiota (host gut, host gut & brine shrimp).** Smaller proportions were shared exclusively with diet (brine shrimp) or with neither the gut microbiota nor diet (other).
(TIF)

**S2 Fig. The proportion of bacterial ASVs of the intestinal tissue that were also detected in fecal samples of the same individual differed substantially within and across individuals and ranged from 21.1–81% (median: 56%).**
(TIF)

**S3 Fig.** Alpha diversity estimates (ASV richness) at different rarefaction depths for different source materials (feces, guts, diet items) of the first (A) and second (B) experiment. The investigated sequencing depths range from 1 to 25,000 reads. We chose 4238 and 4275 reads as the sequencing depths at which the data was rarefied for the first and second experiment, respectively (indicated by vertical lines). At these sampling depths, a large proportion of the microbial diversity is captured across the different sample types.
(TIF)

**S4 Fig.** Across fecal samples, all three alpha diversity measures, ASV richness (A), Shannon diversity (B), and Pielou's evenness (C) were negatively correlated with the overlap between bacterial communities of fecal samples and the gut tissue (based on Spearman's rank correlation coefficient).
(TIF)

**S5 Fig. Pairwise comparisons of Bray-Curtis dissimilarity revealed no significant differences among fecal samples and between fecal and gut samples for all but one individual (Sayward 2).** Higher values indicate stronger dissimilarity among samples. Wilcoxon rank-sum tests, ***$P < 0.001$ (adjusted for multiple comparisons using Bonferroni correction).
(TIF)

**S1 Table. Information on sample type, host population, diet, sampling time point, as well as number of raw and filtered sequencing reads for all samples.**
(XLSX)

**S2 Table. Test statistics of differential abundance analysis between fecal and gut samples in the first experiment using ANCOM in QIIME2 on the phylum level, with and without filtering of diet-derived bacteria.** Test statistics of bacterial phyla that significantly differ between the two tissue types are indicated in bold.
(XLSX)

**S3 Table. Test statistics of differential abundance analysis between fecal and gut samples in the first experiment using ANCOM in QIIME2 on the genus level, with and without filtering of diet-derived bacteria.** Test statistics of bacterial genera that significantly differ between the two tissue types are indicated in bold.
(XLSX)

**S4 Table. Test statistics of differential abundance analysis between fecal samples before and after the diet change in the second experiment using ANCOM in QIIME2 on the phylum level, with and without filtering of diet-derived bacteria.** Test statistics of bacterial phyla that significantly differ between diet treatments are indicated in bold.
(XLSX)

**S5 Table. Test statistics of differential abundance analysis between fecal samples before and after the diet change in the second experiment using ANCOM in QIIME2 on the genus level, with and without filtering of diet-derived bacteria.** Test statistics of bacterial genera that significantly differ between diet treatments are indicated in bold.
(XLSX)

## Author Contributions

**Conceptualization:** Andreas Härer, Diana J. Rennison.

**Data curation:** Andreas Härer.

**Formal analysis:** Andreas Härer.

**Funding acquisition:** Andreas Härer, Diana J. Rennison.

**Investigation:** Andreas Härer, Diana J. Rennison.

**Methodology:** Andreas Härer.

**Project administration:** Diana J. Rennison.

**Resources:** Diana J. Rennison.

**Supervision:** Diana J. Rennison.

**Validation:** Andreas Härer.

**Visualization:** Andreas Härer.

**Writing – original draft:** Andreas Härer.

**Writing – review & editing:** Andreas Härer, Diana J. Rennison.

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
