## [Decision Letter · Decision Letter 0]

11 Jul 2023

PONE-D-23-04495Assessing the validity of fecal sampling for characterizing variation in threespine stickleback’s gut microbiotaPLOS ONE

Dear Dr. Härer,

Thank you for submitting your manuscript to PLOS ONE. After careful consideration, we feel that it has merit but does not fully meet PLOS ONE’s publication criteria as it currently stands. Therefore, we invite you to submit a revised version of the manuscript that addresses the points raised during the review process.

Although two of the three reviewers judged the initial manuscript as requiring only minor revisions, the other reviewer (reviewer 2) cited points about the manuscript that require major revisions, each of which warrants attention. In general, authors must re-analyze their data, and review discussion of results. Some details on the methodology must also be clarified.

We look forward to receiving your revised manuscript.

Kind regards,

Rina Bagsic Opulencia, PhD

Academic Editor

PLOS ONE

Journal Requirements:

"This work was supported by funding from the Deutsche Forschungsgemeinschaft (DFG, German Research Foundation) – project number 458274593 to A.H. and from the University of California San Diego to D.J.R."

"This work was supported by funding from the Deutsche Forschungsgemeinschaft (DFG, German Research Foundation) – project number 458274593 to A.H. and from the University of California San Diego to D.J.R."

"This work was supported by funding from the Deutsche Forschungsgemeinschaft (DFG, German Research Foundation) – project number 458274593 to A.H. and from the University of California San Diego to D.J.R."   

Reviewers' comments:

Reviewer's Responses to Questions

**Comments to the Author**

1. Is the manuscript technically sound, and do the data support the conclusions?

Reviewer #1: Yes

Reviewer #2: Partly

Reviewer #3: Yes

2. Has the statistical analysis been performed appropriately and rigorously? 

Reviewer #1: Yes

Reviewer #2: Yes

Reviewer #3: Yes

3. Have the authors made all data underlying the findings in their manuscript fully available?

Reviewer #1: Yes

Reviewer #2: Yes

Reviewer #3: Yes

4. Is the manuscript presented in an intelligible fashion and written in standard English?

Reviewer #1: Yes

Reviewer #2: Yes

Reviewer #3: Yes

5. Review Comments to the Author

Reviewer #1: This is a well-designed study, and the findings is so important for the wildlife gut microbial research using non-invasive strategies (e.g., feces). Thus, the fecal microbiome could represent the gut microbiome to some extent. I only have one minor suggestions: I am curious on the sample size. For example, line 335-337, Was the small sample size sufficient to capture microbiota structuring?

Reviewer #2: The paper by Harer et al. evaluates the use of feces as an appropriate proxy for intestinal microbiota profiling. The experimental setup is adequate for testing this hypothesis, with a strong point being the inclusion of freshwater and coastal populations of the same fish species, individual fecal samples and the impact of diet manipulation on the microbiota; in the latter, a strong weakness is that there is no data on the gut tissue microbiota impact in order to be compared with the relative feces samples. Despite the scientific interest of this working hypothesis, the paper fails to sound prove some of its claims.

L 52-3, 318-9: It has also been tested in Oncorhynchus mykiss (Mente E, Nikouli E, Antonopoulou E, Martin SAM, Kormas KA (2018) Core vs. diet -associated and postprandial bacterial communities of the rainbow trout (Oncorhynchus mykiss) midgut and feaces. Biology Open 7,bio.034397)

L. 95: at what % of the animals' body size? Frequency of feeding?

L. 111-2: Mechanical forcing to empty the gut from its digesta might leave fecal residues that cannot be detected by the naked eye. Were the fish fasted for 1 day prior to scarifying them? Were the gut samples rinsed before stored? If yes, with what solution?

L. 157-8: this is a rather unfortunate issue and diminishes the phylogenetic attribution of the ASVs but since you targeted only the V4 region (ca. 300bp) it can be marginally accepted.

L. 164-170: I strongly disagree with some of these criteria. Cyanobacteria ASVs might originate from the surrounding water or the feed and for this it is important to know if they make it in gut/feces. Regarding the "more than 10%" I believe this is too arbitrary to amend quantitative data as the readers are interested in all ASVs shared or not between the various habitats you investigated. I suggest to include all ASVs that are left after rarefaction and proceed to your analysis with this dataset.

L. 186: I strongly suggest to also include Simpson 1-D as it varies between 0 and 1 and gives a better picture for evenness, too.

L. 331-3: I do not agree with this statement, as it is. First, feces and gut microbiota were found different in your samples. Second, and most important, you do not distinguish between the two major ecophysiological roles of the gut tissue and the feces microbiota, especially in the frame work of the functional redundancy of bacterial communities. Your work refers ONLY to structural similarities/differences between the gut and the feces but in most cases researchers focus on either habitat for specific reasons.

L. 339-340: you can find some arguments on numbers of replicates in Panteli N, Mastoraki M, Nikouli E, Lazarina M, Antonopoulou E, Kormas KA (2020) Imprinting statistically sound conclusions for gut microbiota in comparative animal studies: A case study with diet and teleost fishes. Comparative Biochemistry and Physiology Part D: Genomics and Proteomics 36:100738

L. 342: this sounds contradictory with L. 331...!

Reviewer #3: I think the manuscript addresses something very important for fish microbiota research, and this is the dynamics of the gut microbial communities, not just due to temporal changes due to feeding but also due to the diet. By establishing non-invasive sampling, like in most terrestrial animal studies, can offer the positibility to study such questions.

Overall the manuscript is well written. I only have some clarifications that I would like to address.

1. Line 101: how were the feces collected? After how many hours from feeding? Did you use a fecal trap? Was the feeding continuous or once and then the authors performed samplings?

2. Line 108: How did the authors removed the residual water? Was it by centrifuge?

3. The authors collected feces by several individuals. It is not clear to me. How do you collect feces per fish? Is there a separation between the individuals? How can you ensure which feces come from which fish if these are kept in nets?

4. Overall comments, the methods section reads a lot like results. So please make a clear separation of where you want to already present data.

6. PLOS authors have the option to publish the peer review history of their article (what does this mean?). If published, this will include your full peer review and any attached files.

Reviewer #1: No

Reviewer #2: No

Reviewer #3: No

---

## [Author Response · Author response to Decision Letter 0]

4 Aug 2023

PONE-D-23-04495

Assessing the validity of fecal sampling for characterizing variation in threespine stickleback’s gut microbiota

PLOS ONE

Dear Dr. Härer,

Thank you for submitting your manuscript to PLOS ONE. After careful consideration, we feel that it has merit but does not fully meet PLOS ONE’s publication criteria as it currently stands. Therefore, we invite you to submit a revised version of the manuscript that addresses the points raised during the review process.

Although two of the three reviewers judged the initial manuscript as requiring only minor revisions, the other reviewer (reviewer 2) cited points about the manuscript that require major revisions, each of which warrants attention. In general, authors must re-analyze their data, and review discussion of results. Some details on the methodology must also be clarified.

We would like to thank the editor for the evaluation of our manuscript and we provide point-by-point responses to the reviewers’ comments below. Please note that the line numbers we provide refer to the version of the manuscript highlighting the changes we made to the original version.

We look forward to receiving your revised manuscript.

Kind regards,

Rina Bagsic Opulencia, PhD

Academic Editor

PLOS ONE

Review Comments to the Author

Reviewer #1:

This is a well-designed study, and the findings is so important for the wildlife gut microbial research using non-invasive strategies (e.g., feces). Thus, the fecal microbiome could represent the gut microbiome to some extent. I only have one minor suggestions: I am curious on the sample size. For example, line 335-337, Was the small sample size sufficient to capture microbiota structuring?

We would like to thank the reviewer for their positive evaluation of our manuscript. We further agree with the reviewer that our sample size was indeed rather small, and for the first experiment it was further limited to three host individuals in one population since PCR amplification failed for the gut sample of one individual from the Sayward population. However, we would like to emphasize that, despite the small sample sizes, we were able to detect significant differences between host populations and between sample types (i.e., feces vs guts). Hence, this gives us confidence that our sample size was sufficient to detect biologically meaningful differences in microbial community composition (lines 359-362). Nonetheless, we understand the reviewer’s concerns and we therefore in the Discussion section recommend having a larger sample size in future studies in order to obtain more robust results, especially in the light of observed variation among samples (lines 362-365). Yet, we still think that our study provides an important contribution, especially since it is the first that investigates temporal variation of fecal microbial communities of individual hosts and can therefore inspire future research.

Reviewer #2: The paper by Harer et al. evaluates the use of feces as an appropriate proxy for intestinal microbiota profiling. The experimental setup is adequate for testing this hypothesis, with a strong point being the inclusion of freshwater and coastal populations of the same fish species, individual fecal samples and the impact of diet manipulation on the microbiota; in the latter, a strong weakness is that there is no data on the gut tissue microbiota impact in order to be compared with the relative feces samples. Despite the scientific interest of this working hypothesis, the paper fails to sound prove some of its claims.

We thank the reviewer for their assessment of our work and the valuable comments that we will address in detail below.

L 52-3, 318-9: It has also been tested in Oncorhynchus mykiss (Mente E, Nikouli E, Antonopoulou E, Martin SAM, Kormas KA (2018) Core vs. diet -associated and postprandial bacterial communities of the rainbow trout (Oncorhynchus mykiss) midgut and feaces. Biology Open 7,bio.034397)

We thank the reviewer for this suggestion and we added this reference to the revised manuscript.

L. 95: at what % of the animals' body size? Frequency of feeding?

We clarified in the Methods section that all fish before and during the experiment were fed once a day (lines 98-99, 124-125). We did not measure the amount of food in relationship to the body size of the fish. However, fish were fed ad libitum and the amount of food was standardized across all experimental fish which we now mention in the revised manuscript (lines 97-99).

L. 111-2: Mechanical forcing to empty the gut from its digesta might leave fecal residues that cannot be detected by the naked eye. Were the fish fasted for 1 day prior to scarifying them? Were the gut samples rinsed before stored? If yes, with what solution?

The reviewer raises a valid point here and while we are confident that we removed the vast majority of fecal matter in the gut, we agree that our method does not ensure that 100% of the digesta is removed. We now acknowledge this technical limitation in the Methods section of our revised manuscript (lines 119-121). The time since last feeding depended on the time point at which a fish was sacrificed, which was either in the morning (24 hours after feeding) or in the afternoon (~6 hours after feeding), which we now mention in the manuscript (line 116). Unfortunately, we did not record the time of day of the last sampling which precludes us from making inferences on whether the time since last feeding might have an effect on the similarity of microbial communities between feces and guts. However, we did not detect evidence for a correlation between overlap in bacterial communities and the time span between fecal sampling and gut sampling meaning that fecal samples collected closer to the gut sampling did not show larger overlap with the gut sample of the same fish. Yet, we acknowledge this limitation in the Discussion section of the revised manuscript (lines 409-412). Lastly, we did not rinse gut samples before we stored them at -80°C.

L. 157-8: this is a rather unfortunate issue and diminishes the phylogenetic attribution of the ASVs but since you targeted only the V4 region (ca. 300bp) it can be marginally accepted.

We agree with the reviewer that using longer sequences is generally better in order to accurately assign taxonomy to the ASVs, however, we would like to highlight that the 250bp we used for our analyses encompass 86% of the target locus (250 out of 291 bp). Thus, we used a large proportion of the sequence information that can be obtained with 16S rRNA sequencing using the established 515F x 806R primers.

L. 164-170: I strongly disagree with some of these criteria. Cyanobacteria ASVs might originate from the surrounding water or the feed and for this it is important to know if they make it in gut/feces. Regarding the "more than 10%" I believe this is too arbitrary to amend quantitative data as the readers are interested in all ASVs shared or not between the various habitats you investigated. I suggest to include all ASVs that are left after rarefaction and proceed to your analysis with this dataset.

We thank the reviewer for this critical comment and we would like to explain why we chose to exclude reads belonging to different microbes. First, we removed Cyanobacteria since these are commonly free-living photosynthetic bacteria that have also been filtered from fish gut microbiomes in some previous studies (e.g., Bolnick et al. 2014 Nat Comm, Baldo et al. 2017 ISMEJ), while other studies have included them in their analyses. Our main reason for filtering Cyanobacteria was that we wanted to exclude free-living bacteria that we did not expect to be part of the gut microbiota in order to investigate bacteria that are actually permanent and functional members of the gut microbiota rather than bacteria that are acquired from the environment. We understand that this might actually be true for other bacteria as well but Cyanobacteria are an obvious group that are not expected to inhabit the fish gut. Before filtering Cyanobacteria, we checked for their abundance and distribution across samples. We found that they only comprised an average of 1.5% of all sequencing reads across our samples whereas only 17 out of 94 samples showed a proportion of more than 1% and 60 samples did not contain any Cyanobacteria at all. Further, only one of the four diet samples contained any Cyanobacteria with a very low proportion of 0.06%. This uneven distribution and lack of Cyanobacteria in the majority of samples, in our opinion, provides evidence that they are indeed environmental microbes that were passively acquired and therefore not relevant to the main question of our study. Unfortunately, as we mention in our manuscript, we did not sample eDNA from the tank water and, thus, we are not able to characterize the community of free-living bacteria but we agree that it would be interesting to obtain a more detailed picture of which bacteria are associated with the water, diet, and fish guts and what proportion of these bacteria are shared among environments/sample types. We mention this point in the Discussion section of our manuscript (lines 413-420).

However, in response to the reviewer’s suggestion we tested whether our results and conclusions would change if cyanobacteria were included. Overall, we found qualitatively similar results for all of our alpha and beta diversity analyses which gives us confidence that our conclusions are robust and not affected by the presence/absence of Cyanobacteria in our data. The only exception was for PERMANOVA tests of weighted UniFrac measures where we found significant effects of host population and sample type with p-values of 0.028 and 0.039 after filtering of Cyanobacteria whereas evidence was only suggestive including cyanobacteria with p-values of 0.069 and 0.077. Yet, the r-squared values were very similar and since all other results did not differ, we conclude that filtering Cyanobacteria did not affect our results. However, we now mention our justification for excluding Cyanobacteria in the Methods section and highlight that this filtering step did not substantially affect our results (lines 177-182). We further added R scripts and alpha and beta diversity data files without the Cyanobacteria-filtering step to the figshare repository.

Regarding the second point on filtering sequencing reads that were highly abundant in diet items, we would first like to highlight that this led only to the exclusion of four bacterial ASVs which we now mention in the revised Methods section (lines 186-192). We chose the threshold of 10% as a compromise between excluding the ASVs that comprised the majority of the diet-associated bacteria while excluding as few ASVs as possible. However, we know that the threshold of 10% does not represent a fixed number and was chosen by us based on the information on the microbial community composition of our data. Regardless, we definitely agree with the reviewer that it is important to understand if and how this filtering step might affect our results and conclusions and, thus, we performed all statistical analyses with and without filtering these four bacterial ASVs. The results were overall qualitatively similar and we mention in the manuscript when results differed. We further provide R scripts and alpha and beta diversity data files with and without the diet-associated filtering step in the figshare repository. In our opinion, including both datasets strengthens the main results of our manuscript and the diet-associated bacterial filtering step is crucial to our conclusion that the differences between fecal samples of fish fed different diets is not merely a result of the differential acquisition of diet-associated bacteria. We now highlight this in the Discussion sections of our revised manuscript (lines 448-450).

L. 186: I strongly suggest to also include Simpson 1-D as it varies between 0 and 1 and gives a better picture for evenness, too.

We thank the reviewer for this comment and we agree that adding an additional alpha diversity metric calculating evenness might strengthen our results. We decided to include the Pielou’s evenness measure rather than the suggested Simpson index since it also represents a measure of evenness with values ranging from 0 to 1 and it is further a part of the QIIME2 workflow that we used for our data analysis. We found that results were largely qualitatively similar across all three alpha diversity metrics and we added the results for Pielou’s evenness to the revised manuscript.

L. 331-3: I do not agree with this statement, as it is. First, feces and gut microbiota were found different in your samples. Second, and most important, you do not distinguish between the two major ecophysiological roles of the gut tissue and the feces microbiota, especially in the frame work of the functional redundancy of bacterial communities. Your work refers ONLY to structural similarities/differences between the gut and the feces but in most cases researchers focus on either habitat for specific reasons.

We agree with the reviewer that we only investigated the taxonomic composition of bacterial communities based on 16S rRNA sequencing but we did not study the functional metagenome. Thus, we are limited to making inferences about similarities and differences in microbiota composition. While we think that we found some evidence that fecal samples can be used as a proxy for the gut microbiota in stickleback (despite the differences we observed between sample types that we acknowledge in the Discussion section), we understand that we might have not presented the point we are trying to make completely clearly. As we argue in the Discussion section (lines 358-362), we found that host population explained twice as much of the variation in microbiota composition than sample type (feces vs guts). Further, the major bacterial phyla were shared between feces and guts whereas the diet change led to substantial changes in bacterial community composition of fecal samples (lines 372-383). In our opinion, these results show that we were able to pick up differences in microbiota composition between host populations which provides biologically meaningful results.

We agree with the reviewer that we cannot make functional inferences (and we were not trying to) but we would also like to highlight that the question of whether microbial communities obtained from fecal samples reflect the gut microbiota has been asked in many study systems. For many vertebrate species and, especially species of conservation concern, fecal samples are often used as proxies for studying gut microbiota variation across hosts. Our study adds to this literature using threespine stickleback, a famous model system in evolutionary ecology whose gut microbiota has been investigated in many studies over the last decade, while also being the first one to study temporal variation in microbial community composition of the same individual in fish. We overall think that we found substantial evidence that fecal samples could be used as proxies for the gut microbiota in stickleback, while also discussing the limitations of our data. In response to the reviewer’s comment, we changed the phrasing of our general conclusion in the Discussion section as well as in the abstract (lines 28-30, 491-495) to better reflect that we found conflicting evidence regarding the suitability of fecal samples while still making our case in the Discussion section of why we think fecal samples could be suitable for gut microbiota studies in stickleback. We would further like to emphasize that even if the microbial communities associated with fecal samples do not perfectly reflect the gut microbiota (which is a common observation across study systems), we clearly show that fecal samples allow detecting differences between host populations and diet treatments, making them a promising tool for experimental manipulation of the gut microbiota with non-lethal sampling. We would further like to highlight that our study is the first that investigated temporal variation in gut microbiota composition in individual fish, which provides crucial insights into the suitability of fecal sampling as a proxy for studying the gut microbiota.

L. 339-340: you can find some arguments on numbers of replicates in Panteli N, Mastoraki M, Nikouli E, Lazarina M, Antonopoulou E, Kormas KA (2020) Imprinting statistically sound conclusions for gut microbiota in comparative animal studies: A case study with diet and teleost fishes. Comparative Biochemistry and Physiology Part D: Genomics and Proteomics 36:100738

We thank the reviewer for providing this reference and we added it to our manuscript. We also now more clearly highlight that while we found significant differences between sample types and host populations, larger sample sizes would be necessary in order to draw more robust conclusions (lines 363-367) and we also acknowledge the limitations of our rather small sample sizes throughout the Discussion section.

L. 342: this sounds contradictory with L. 331...!

In response to a comment above, we rephrased the beginning of this paragraph to highlight that we indeed found evidence for and against the validity of fecal samples as a proxy for the gut microbiota. The section referenced by the reviewer represents one of the arguments that the bacterial communities of fecal and gut samples are actually not that different and we specifically refer to the results of Figure S5 here. So yes, the fact that we found overall differences in bacterial community composition between sample types across host individuals based on PERMANOVA tests provides a different conclusion than the analysis used to produce Figure S5, which measure the differences in beta diversity distances among fecal samples and between fecal and gut samples of the same individual. In sum, these represent distinct analyses that investigate different aspects and lead to different conclusions that argue for and against fecal sampling as a good proxy, and we tried to clarify that we indeed found arguments for both sides (lines 354-357, 370-372).

Reviewer #3: I think the manuscript addresses something very important for fish microbiota research, and this is the dynamics of the gut microbial communities, not just due to temporal changes due to feeding but also due to the diet. By establishing non-invasive sampling, like in most terrestrial animal studies, can offer the possibility to study such questions.

Overall the manuscript is well written. I only have some clarifications that I would like to address.

We thank the reviewer for their positive feedback and we will clarify the points raised below.

1. Line 101: how were the feces collected? After how many hours from feeding? Did you use a fecal trap? Was the feeding continuous or once and then the authors performed samplings?

We clarified our description of the experimental set-up (lines 98-100, 102-105, 115-116). The fish were fed once a day (in the morning) and we checked for feces twice a day, right before feeding or 6 hours after feeding. Feces were collected with disposable plastic pipettes. All these details can be found in the revised Methods section of our manuscript under “Experimental set-up and data collection”.

2. Line 108: How did the authors removed the residual water? Was it by centrifuge?

We actually removed the residual water by pipetting and we added this information to the revised manuscript (line 115).

3. The authors collected feces by several individuals. It is not clear to me. How do you collect feces per fish? Is there a separation between the individuals? How can you ensure which feces come from which fish if these are kept in nets?

Each fish was kept in a separate fine-meshed net breeder that was not permeable to feces. Hence, this allowed us to collect fecal samples for each individual fish from the bottom of the net breeder using a disposable plastic pipette. We added information on this procedure to the revised Methods section (lines 103-104).

4. Overall comments, the methods section reads a lot like results. So please make a clear separation of where you want to already present data.

We thank the reviewer for this comment and we tried to clarify where in the Methods section we present results of our study; however, we were not sure which exact parts the reviewer referred to. Specifically, we clarified that the finding that fecal samples collected in the morning or afternoon did not differ in their alpha- or beta-diversity is part of our results (lines 129-131). However, since this result is rather technical, we decided to keep it in the Methods section but if the reviewer has a strong opinion about moving this to the Results section, we’d be happy to do so.

---

## [Decision Letter · Decision Letter 1]

18 Aug 2023

Assessing the validity of fecal sampling for characterizing variation in threespine stickleback’s gut microbiota

PONE-D-23-04495R1

Dear Dr. Härer,

We’re pleased to inform you that your manuscript has been judged scientifically suitable for publication and will be formally accepted for publication once it meets all outstanding technical requirements.

Kind regards,

Rina Bagsic Opulencia, PhD

Academic Editor

PLOS ONE

Additional Editor Comments (optional):

Reviewers' comments:

Reviewer's Responses to Questions

**Comments to the Author**

1. If the authors have adequately addressed your comments raised in a previous round of review and you feel that this manuscript is now acceptable for publication, you may indicate that here to bypass the “Comments to the Author” section, enter your conflict of interest statement in the “Confidential to Editor” section, and submit your "Accept" recommendation.

Reviewer #1: All comments have been addressed

Reviewer #2: All comments have been addressed

Reviewer #3: All comments have been addressed

2. Is the manuscript technically sound, and do the data support the conclusions?

Reviewer #1: Yes

Reviewer #2: Yes

Reviewer #3: Yes

3. Has the statistical analysis been performed appropriately and rigorously? 

Reviewer #1: Yes

Reviewer #2: Yes

Reviewer #3: Yes

4. Have the authors made all data underlying the findings in their manuscript fully available?

Reviewer #1: Yes

Reviewer #2: Yes

Reviewer #3: (No Response)

5. Is the manuscript presented in an intelligible fashion and written in standard English?

Reviewer #1: Yes

Reviewer #2: Yes

Reviewer #3: Yes

6. Review Comments to the Author

Reviewer #1: I have read this manuscript again, and find that the authors have addressed the comments in this revision. I have no comments now.

Reviewer #2: The authors have recognised the limitations raised in mt first round of review and they have satisfactory handled them. This is a very honest approach to be found in a published paper and as the paper is very interesting and well-designed and performed, I agree for its acceptance.

Reviewer #3: (No Response)

7. PLOS authors have the option to publish the peer review history of their article (what does this mean?). If published, this will include your full peer review and any attached files.

Reviewer #1: No

Reviewer #2: No

Reviewer #3: No

---

## [Editor Report · Acceptance letter]

15 Sep 2023

PONE-D-23-04495R1 

Assessing the validity of fecal sampling for characterizing variation in threespine stickleback’s gut microbiota 

Dear Dr. Härer:

I'm pleased to inform you that your manuscript has been deemed suitable for publication in PLOS ONE. Congratulations! Your manuscript is now with our production department. 

Kind regards, 

on behalf of

Dr. Rina Bagsic Opulencia 

Academic Editor

PLOS ONE